# Magnetism and EPR Spectroscopy of Nanocrystalline and Amorphous TiO₂: Fe upon Al Doping

Anatoly Yermakov [1,*] , Mikhail Uimin [1], Kirill Borodin [1,2], Artem Minin [1,2,*] , Danil Boukhvalov [2,3] ,
Denis Starichenko [1] , Alexey Volegov [1,2], Rushana Eremina [4], Ivan Yatsyk [4] , Galina Zakharova [5]
and Vasiliy Gaviko [1,2]

1   M.N. Mikheev Institute of Metal Physics, Ural Branch of the Russian Academy of Sciences,
    620108 Yekaterinburg, Russia
2   Institute of Natural Sciences and Mathematics, Ural Federal University, 620002 Yekaterinburg, Russia
3   College of Science, Institute of Materials Physics and Chemistry, Nanjing Forestry University,
    Nanjing 210037, China
4   Zavoisky Physical-Technical Institute, Federal Research Center "Kazan Scientific Center of RAS",
    420029 Kazan, Russia
5   Institute of Solid State Chemistry, Ural Branch of the Russian Academy of Sciences,
    620990 Yekaterinburg, Russia
*   Correspondence: yermakov.anatoly@gmail.com (A.Y.); calamatica@gmail.com (A.M.)

**Abstract:** This work is devoted to the study of the magnetic properties and Electron Paramagnetic Resonance (EPR) spectroscopy of TiO₂:Fe nanoparticles doped with Al in different structural states. The sol-gel methods have been used to obtain the particles in both crystalline (average size from 3 to 20 nm) and X-ray amorphous states. The electron paramagnetic resonance spectra of crystalline samples TiO₂:Fe doped with aluminum besides a resonance line with g-factor ~2 exhibit a small signal with a g-factor of 4.3 from $Fe^{3+}$ ions with rhombohedral distortions. The fraction of $Fe^{3+}$ with rhombohedral distortions increases with increasing aluminum content. For the amorphous state at Al doping, the resonance with a g-factor of 4.3 is completely dominant in the electron paramagnetic resonance spectrum. The density functional theory calculation shows that aluminum prefers to be localized near iron ions, distorting the nearest $Fe^{3+}$ environment. The complex integral electron paramagnetic resonance spectrum of all samples was fitted with sufficient accuracy by three separate resonance lines with different widths and intensities. The temperature behavior of the electron paramagnetic resonance spectrum can be described by the coexistence of paramagnetic centers (isolated $Fe^{3+}$ ions including dipole-dipole interactions) and iron clusters with negative exchange interactions.

**Keywords:** TiO₂:Fe doped by Al; nanocrystalline state; amorphous state; magnetism; EPR; DFT calculation; Fe and Al localization

## 1. Introduction

Titanium dioxide is a widespread semiconductor that is applied in various fields, mainly as a photocatalyst for various chemical reactions [1–7]. In a number of works, special attention is paid to the morphological and microstructural features of TiO₂ nanopowders doped with trivalent Al for various, predominantly photocatalytic, applications [8–16]. It should be emphasized that the authors of the listed works do not always pay attention to the fundamental point regarding Al localization in the lattice and in no way associate the high catalytic activity with the electronic properties of Al and TiO₂.

Potentially, titanium dioxide doped with 3d metals can be used for spintronic purposes [8,17]. In this regard, the nature of magnetism in the compound doped with 3d ions is of particular interest [18–22]. There are several works that have discussed the magnetism of systems based on TiO₂ doped with Al [23]. The results demonstrate that the adsorption

of Al on the surfaces of $TiO_2$ nanostructures yields spontaneous magnetization. Based on experiments and DFT simulations, the authors explain the emergence of RT ferromagnetism because of charge-transfer from Al to O atoms [23,24].

As a critical analysis of the mentioned works, it is worth noting that there are numerous contradictory data in the literature, both on the localization of Al in the $TiO_2$ lattice and on the defect state of $TiO_2$ doped with Al.

To develop more efficient materials for various purposes based on $TiO_2$ doped with Al and Fe (catalysts, sensors, solar cells, spin electronics, etc.), it is desirable to answer the several critical questions formulated below.

How do the magnetic and electronic properties of nanopowders change upon the doping of aluminum into $TiO_2$:Fe prepared by sol-gel methods in different structural states (crystalline or amorphous)? What sites will aluminum prefer to occupy in the $TiO_2$:Fe lattice with an anatase structure, and will they form a substitutional or interstitial solution in the $TiO_2$ lattice? Will the valence state of titanium change upon the introduction of aluminum? What is the role of vacancies, for instance, upon an isovalent substitution of $Fe^{3+}$ by $Al^{3+}$ in the formation of the magnetic properties of $TiO_2$ with dopants? How can the exchange interactions in $Fe^{3+}$ clusters or (dimers) change upon doping with Al? Will clustering of aluminum in the $TiO_2$ lattice be observed?

In this paper, an attempt is made to obtain answers to some of the abovementioned important questions.

Magnetic and electron paramagnetic resonance (EPR) methods are very sensitive to the electronic state and distortions near the iron ion when $TiO_2$:Fe is doped with aluminum. The $Fe^{3+}$ ion as a carrier of the magnetic moment and as an EPR probe can provide detailed information on both the localization of aluminum in the lattice and the change in the electronic state of the Fe ion in terms of the magnetization of the $TiO_2$:Fe system upon doping with Al. As far as we know, the case of the binary co-doping (Fe and Al) of $TiO_2$ has not been investigated in this respect. In this work, a detailed study of the magnetic and resonance properties of a $TiO_2$-based system doped with iron and aluminum has been carried out.

It was of interest to study whether there will be any differences in the magnetic properties and EPR spectra of samples with binary doping when dopants are distributed in crystalline or amorphous $TiO_2$ matrices.

The choice of samples with a quasi-amorphous structure and the comparison with samples with a more perfect crystal structure of anatase have the following motivation. In some approximation, the X-ray amorphous phase based on $TiO_2$ can be structurally modeled as a glass state with a different set of atoms' distribution in the nearest coordination sphere. Presumably, in the X-ray amorphous phase, iron and aluminum ions with a high probability can occupy sites that do not occur in a perfect crystal lattice of anatase. It can be expected that there is a wider set of short-range order (SRO), which differs from the short-range order in the ordered structure of anatase. In this regard, the search for differences in the magnetic and EPR properties of $TiO_2$:Fe upon doping the X-ray amorphous and crystalline phases of anatase with aluminum is of fundamental interest.

The nature of EPR resonance (in particular a line with a g-factor of about 4.3) is sometimes associated with the existence of iron surface states in nanoparticles or $Fe^{3+}$ sites with local rhombohedral distortions in a titanium dioxide lattice or the existence of tetrahedral environments. To clarify the most energetically favorable configurations of iron and aluminum impurities in a titanium dioxide matrix with an anatase structure, a series of ab initio calculations was performed. The calculations were carried out within the framework of the theory of the electron density functional theory (DFT) using the pseudopotential computer code SIESTA [25].

## 2. Materials and Methods

### 2.1. Synthesis of the Crystalline TiO$_2$:Fe, Al Nanopowders (the First Chemical Route)

Aluminum-doped anatase TiO$_2$ nanopowders with iron impurities were prepared using the autoclave-assisted hydrothermal method [26]. The following substances were used as starting materials: a 15% solution of titanium chloride (III, Merck, Darmstadt, Germany) in a 10% HCl solution, and "chemically pure" anhydrous aluminum nitrate and ammonium hydroxide. The samples were synthesized as follows: Al(NO$_3$)$_3$·9H$_2$O was dissolved in TiCl$_3$ according to a predetermined molar ratio, then NH$_4$OH was added dropwise under constant stirring until pH = 9.2 was reached. The reaction mass was placed in an autoclave at 433 K for 24 h, and then cooled down to room temperature under atmospheric conditions. The resulting product was filtered, washed with water to neutralize the powder, and dried in air at 320 K. Undoped titanium dioxide nanopowder was synthesized using the same method as the Al-doped samples but without adding anhydrous aluminum nitrate.

Synthesis of TiO$_2$:Fe, Al samples with X-ray amorphous state, including a small size of nanoparticles (the second chemical route): As a second variant of the synthesis of titanium oxide nanoparticles doped with iron and aluminum, the method of acid hydrolysis of organometallic precursors was used. Titanium isopropyl oxide (Titanium (IV) isopropoxide) was used for the synthesis; iron (III) acetylacetonate and aluminum acetylacetonate (aluminum acetylacetonate) were used as a source of metal ions. All reagents used were obtained from Sigma Aldrich, St. Louis, MO, USA. All work tools were made of non-ferromagnetic materials (plastic, titanium) and were treated with 2 M hydrochloric acid to minimize ferromagnetic contamination.

The synthesis technique was based on the method which allows achievement of the maximum inclusion of metal ions in the structure of titanium oxide due to the preliminary formation of mixed organometallic complexes [27]. The preliminary required amount of titanium isopropoxide (2 mL), iron and aluminum acetylacetonate was placed in a 5 mL test tube, after which 2 mL of acetone was poured into it. After vigorous shaking, a yellowish-brown solution was formed due to iron ions. The solution was left overnight. Then, 1 mL of a 0.1 M aqueous solution of hydrochloric acid was poured into the solution. The hydrolysis reaction occurred almost instantly, however, so that the process runs uniformly throughout, and the entire volume of the colloid is subjected to intense ultrasonic action using an ultrasonic activator with a submerged titanium probe. Ultrasound exposure was performed for 30 s.

Then, the particles were separated by centrifugation (10,000 RPM) for 10 min and were washed with acetone. The washing process was repeated three times. The resulting precipitate was dried at 70 °C. The precursor was further calcined in air at different temperatures (from 300 to 500 °C).

The chemical composition of the synthesized powders was determined using inductively coupled plasma mass spectrometry (ICP-MS) using a Spectromass 2000 spectrometer (Spectro Analytical Instruments, Fitchburg, MA, USA) and X-ray fluorescence analysis. The specific surface area was measured using the BET method.

### 2.2. Samples

Thus, to obtain nanopowders in various structural states, the synthesis was carried out by two chemical routes (See Section 2.1). The first route (CR1) makes it possible to obtain crystalline TiO$_2$:Fe, Al nanopowders with more perfect anatase structure. Another route for sample preparation with tiny nanoparticles TiO$_2$:Fe, Al and even X-ray amorphous state was performed with a different chemical procedure (CR2). These samples in the text and in Table 1 of the article will be designated as CR1 (S1) and CR2 (S2).

**Table 1.** The composition and the average particle size of the samples S1 and S2 sets. S2 samples were calcinated at 300 and 500 °C. Particle sizes were estimated using the BET method.

| Sample | Chemical Route | $T_{calc.}$ | Notation | Composition, at.% | | Size, nm |
|---|---|---|---|---|---|---|
| | | | | Fe | Al | |
| $TiO_2$:Fe | CR1 | - | F-S1 | 1.57 | - | 21 |
| $TiO_2$:Al | CR 1 | - | A-S1 | - | 3.48 | 23 |
| $TiO_2$:FeAl | CR 1 | - | FA-S1 | 1.16 | 3.49 | 16 |
| $TiO_2$:FeAl | CR 2 | 300 | FA-S2-1-300 | 0.90 | 0.04 | 4 |
| $TiO_2$:FeAl | CR 2 | 300 | FA-S2-2-300 | 0.96 | 0.81 | 4 |
| $TiO_2$:FeAl | CR 2 | 300 | FA-S2-3-300 | 0.83 | 1.4 | 3 |
| $TiO_2$:FeAl | CR 2 | 500 | FA-S2-1-500 | 0.96 | 0.012 | 19 |
| $TiO_2$:FeAl | CR 2 | 500 | FA-S2-2-500 | 0.72 | 1.11 | 17 |
| $TiO_2$:FeAl | CR 2 | 500 | FA-S2-3-500 | 1.0 | 1.96 | 13 |

### 2.3. Structural Characterization

X-ray diffraction studies were carried out in copper radiation by means of an Empyrean Series 2 diffractometer manufactured by PANalytical, (Malvern, UK), with Cu Kα radiation. To calculate the size of coherent scattering regions (CSR) and phase compositions, the HighScore v.4.x software (PANalytical, Malvern, UK) package was used.

To analyze microstructure of samples the transmission electron microscope Tecnai G2 30 (FEI Company, Hillsbro, OR, USA) with maximum accelerating voltage up to 300 kV was used.

### 2.4. Magnetic Measurement

The magnetic properties of $TiO_2$:Fe nanopowders were analyzed using Faraday balance at room temperature in fields of up to 12 kOe and a SQUID-magnetometer Quantum Design MPMS XL7 (Quantum Design, San Diego, CA, USA) a wide range of fields (up to 70 kOe) and temperature range (2–300 K). The temperature dependencies of the magnetic susceptibility were measured using field cooled protocols.

### 2.5. Spectroscopic Investigations

Room temperature EPR measurements were performed on a standard homodyne X-band (9.4 GHz) Adani CMS 8400 spectrometer (Adani, Minsk, Belarus). For low temperature (8–170 K) investigations, the Bruker ELEXSYS E580 FT/CW (Bruker Optik GmbH, Ettlingen, Germany) in the range 300–6300 Oe was used under conditions for the field sweep of $\delta Bsw \geq 10\Delta B$ ($\Delta B$ is the peak-to-peak EPR linewidth of the total spectrum); microwave power was 4.7 mW, the modulation amplitude $-1$ Oe. The sample powder was placed in a special quartz tube 4.0 mm in diameter. The sample volume ranged from 2.0 to 5.0 mm$^3$. The spectra were recorded with a super high-Q rectangular resonator (Bruker Optik GmbH, Ettlingen, Germany) Absolute values of the g-factors were calibrated by using chromium chloride $CrCl_3$ (S = 3/2) with g = 1.986. All spectra presented in the figures are normalized on the sample mass and presented in arbitrary units.

### 2.6. Computational Details

Calculations within the framework of the electron density functional theory (DFT) were carried out using the pseudopotential computer code SIESTA. All calculations were carried out using the version of the Purdew–Burk–Ernzerhof generalized gradient approximation (GGA-PBE) for the exchange correlation potential [28]. A complete optimization of atomic positions was carried out, during which the electronic ground state was sequentially found using normalizing pseudopotentials for nuclei and a double plus polarization basis of localized orbitals for all atoms. Forces and total energies were optimized with an

accuracy of 0.04 eV/Å and 1.0 MeV, respectively. The calculations were performed for a $2 \times 2 \times 2$ titanium dioxide supercell (96 atoms). Formation energies were calculated by the standard formula:

$$E_{form} = \{E(host + mFe) - [E(host) + mE(Fe) - mE(Ti)]\}/m,$$

where E(host) and E(host + mFe) are the total energies of the $TiO_2$ supercell with and without oxygen vacancies and with interstitial Al-impurity in the interstitial void (see Figure 1) before and after incorporation of m substitutional iron impurities, and E(Fe) and E(Ti) are the total energies per atom of alpha-Fe and alpha-Ti, respectively.

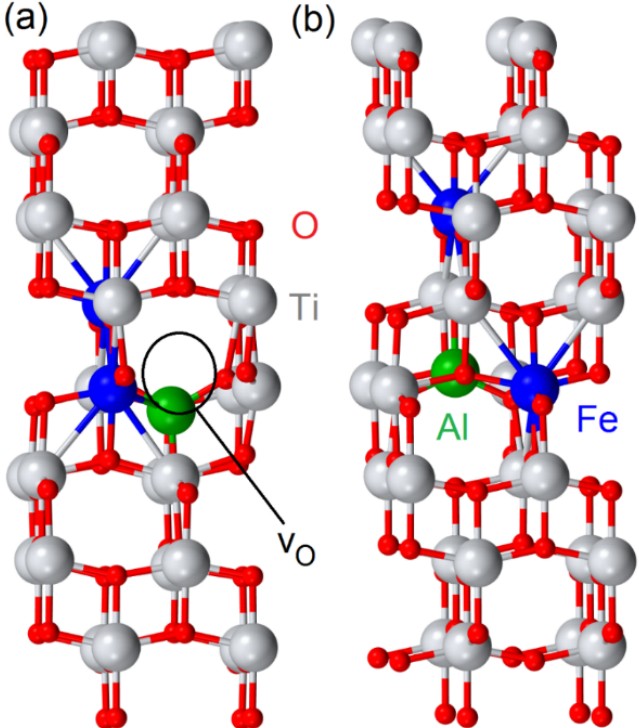

**Figure 1.** Optimized atomic structures of the most energetically favorable configurations of substitutional Fe-impurities in the vicinity of Al-impurity in interstitial void in $TiO_2$ matrix with (**a**) and without (**b**) oxygen vacancy.

## 3. Results

### 3.1. Calculation of Energetically Favorable Configurations of Iron and Aluminum Impurities in the $TiO_2$ Matrix

To search for the most energetically favorable configurations of iron and aluminum impurities in the titanium dioxide matrix, a series of ab initio calculations was performed. This has been used successfully in the past for the study of impurities in the bulk of nanocrystalline oxides and on their surfaces [29].

The first step of our modeling was to calculate the formation energy of a single aluminum impurity for ideal titanium dioxide and in the presence of and without an oxygen vacancy (Figure 1a,b). For the substitutional impurity, the calculated formation energies turned out to be +0.99 and −1.49 eV, and for the interstitial impurity, −3.99 and −3.32 eV. Thus, substitutional aluminum impurities can be excluded from further consideration.

The next step of our modeling was calculations for a single iron impurity in the presence of interstitial aluminum. Results of the calculations demonstrate the visible tendency of the segregation of substitutional Fe-impurities in the first three coordination spheres around the interstitial Al-impurity (see dashed Figure 2A). The presence of the oxygen vacancies insignificantly influences an overall pattern of the segregation of single

Fe impurities in the vicinity of the interstitial Al-center. To evaluate the influence of the interstitial Al-impurity on Fe-Fe interactions, we considered a pair of substitutional iron impurities located at different distances. Results of the calculations (see solid lines in Figure 2) demonstrate that, in this case, the tendency of iron impurities to segregate is less distinct than in the case of a single iron impurity. Note that the formation energies per impurity are the same for 2Fe(S) + Al(I) and Fe(S) + Al(I) configurations. Similarly, in the case of a single iron impurity, the presence of oxygen vacancies does not influence the pattern of impurities' distribution.

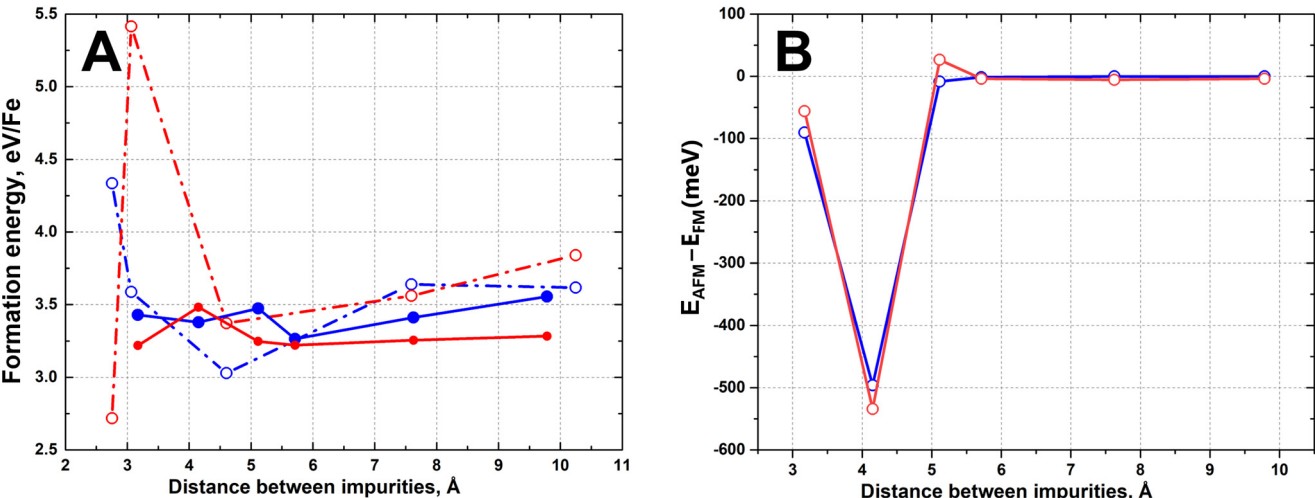

**Figure 2.** Formation energies (**A**) as a function of the distance between aluminum in the interstitial site and iron at the site (thin dashed lines) and two iron impurities at the sites, one of which is located near aluminum in the interstitial site (thick solid lines), and the difference between the total energies of ferro- and antiferromagnetic configurations (**B**) for these pairs. Blue corresponds to impurities in an ideal crystal, red to crystals with an oxygen vacancy near an aluminum impurity.

To evaluate the magnetic properties of various configurations of the defects, we performed the calculations of the total energies of the systems with pairs of iron impurities for antiparallel (AFM) and parallel (FM) orientations of the spins on iron atoms. The negative sign in this case corresponds with the favorability of the antiparallel orientation of the spins on magnetic centers. Since the formation energies of various Fe-Fe pairs is rather close, we performed the described calculations for all Fe-Fe configurations. Results of the calculations (Figure 2B) demonstrate that the ground state magnetic configuration of the studied systems is antiferromagnetic for all Fe-Fe configurations except one. In the case of remote iron impurities, the difference between the values of total energies of the systems with parallel and antiparallel orientations of spins is in the order −1~−5 meV. When the distance between iron impurities decreased, the magnitude of the antiferromagnetic exchange interactions increased. The difference between the total energies of different orientations of the spins for these configurations is in the order of −50~−500 me, which corresponds with robust anti-ferromagnetism.

Thus, the results of theoretical modelling demonstrate the presence in $TiO_2$ co-doped with aluminum and iron of Fe-Fe pairs with robust ferromagnetism, and a significant amount of iron impurities form the pairs with weak antiferromagnetic interactions. Additionally, a visible part of the iron impurities will not participate in the formation of antiferromagnetic pairs or clusters and a tiny number of impurities can form Fe-Fe pairs with weak ferromagnetic interactions.

Thus, based on the results of the calculations, we can conclude that, in $TiO_2$ co-doped with iron and aluminum, iron impurities will be segregated around aluminum centers with the formation of Fe-Fe pairs but a visible amount of iron impurities is distributed as

single impurities in the vicinity of aluminum centers or oxygen vacancies. Thus, all these configurations should be considered in the description of magnetic properties.

The experimental results and their correspondence with the conclusions based on the ab initio calculations are presented and analyzed below.

### 3.2. Structural Characterization

X-ray Characterization of the Samples

Figure 3 shows the X-ray diffraction patterns of samples FS-1 and FAS-1. They contain the reflections of one phase, anatase, and the lattice parameters do not differ from the bulk data within the error. The size of the region of coherent scattering (CSR), estimated from the width of the reflections is, on average, 20 nm.

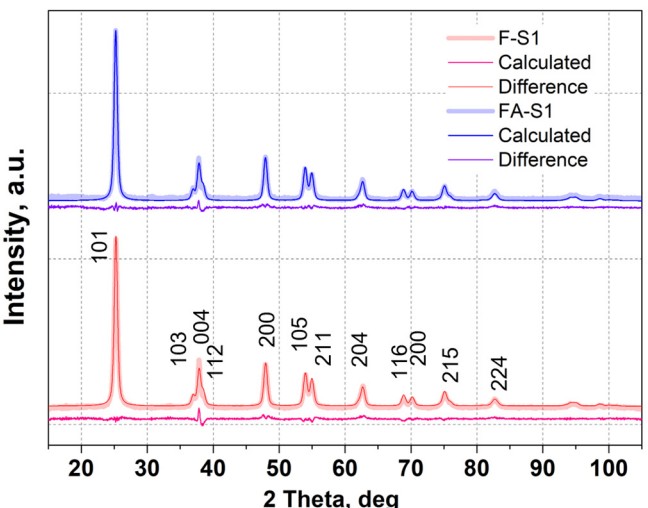

**Figure 3.** X-ray diffraction patterns of the samples F-S1 and FA-S1 with Rietveld refinement.

For samples of the CR2 series calcined at 300 °C, the diffraction lines are much wider, and at a high content of Al, the structure can be identified as X-ray amorphous (Figure 4). Weak reflections near the angles of 2Theta~32° for samples 1 and 2 can be attributed to the brookite phase. A CSR size of FAS2-1-300 and FAS2-2-300 samples is about 4 nm.

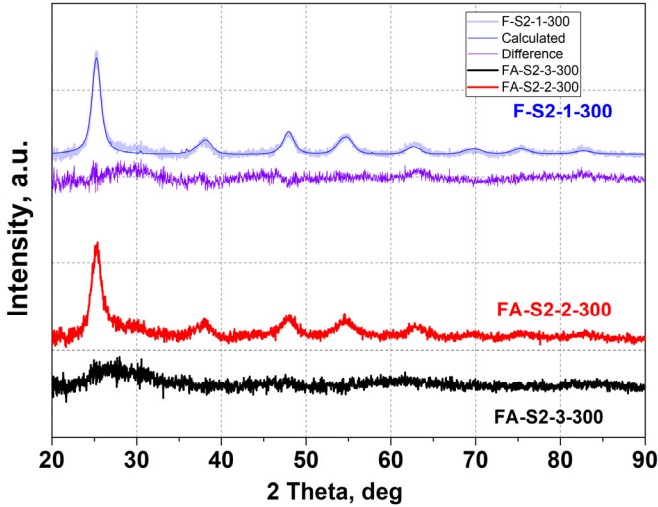

**Figure 4.** X-ray diffraction patterns of samples: (1) FA-S2-1-300, (2) FA-S2-2-300, (3) FA-S2-3-300 with Rietveld refinement.

### 3.3. TEM Investigation

Transmission electron microscopy of the FA-S2-3-300 and FA-S2-3-500 samples shows different patterns depending on the calcination temperature. Calcination at 300 °C leads to the formation of an amorphous state (amorphous halo in the electron diffraction pattern) with a weak contrast at bright-field TEM observation with practically indistinguishable imperfect nanoparticles with sizes of less than 3–5 nm (Figure 5a), in accordance with the results of X-ray analysis and BET. After calcination at 500 °C, the nanoparticles enlarge their sizes by more than 10 nm. At the direct TEM resolution, the atomic planes of the anatase lattice in individual particles are clearly visible (Figure 5c). The electron diffraction pattern (Figure 5d) corresponds to the anatase structure.

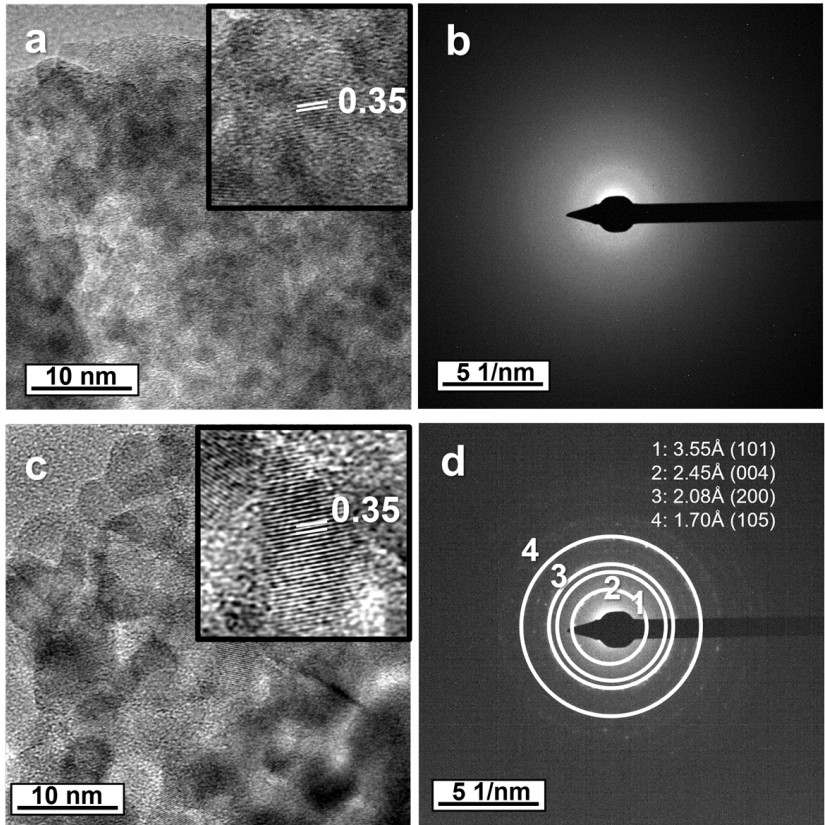

**Figure 5.** Bright-field images (**a**,**c**) and electron diffraction patterns (**b**,**d**) of FA-S2-3-300 sample (**a**,**b**) and FA-S2-3-500 sample (**c**,**d**).

### 3.4. Magnetic Properties

Figure 6 shows the magnetization curves of FS1 and FAS1 samples at 2 K. A significant difference is observed between the experimental magnetization curves of both samples from the Brillouin function (see Figure 6), which should take place for non-interacting paramagnetic centers. Earlier, this difference was studied in detail in the $TiO_2$:Fe system and it was explained by the presence of clusters with negative exchange interactions between iron ions [30]. Only a relatively small change in the effective moment per iron atom can be observed for sample FAS1 containing aluminum, compared to sample FS1 containing no aluminum.

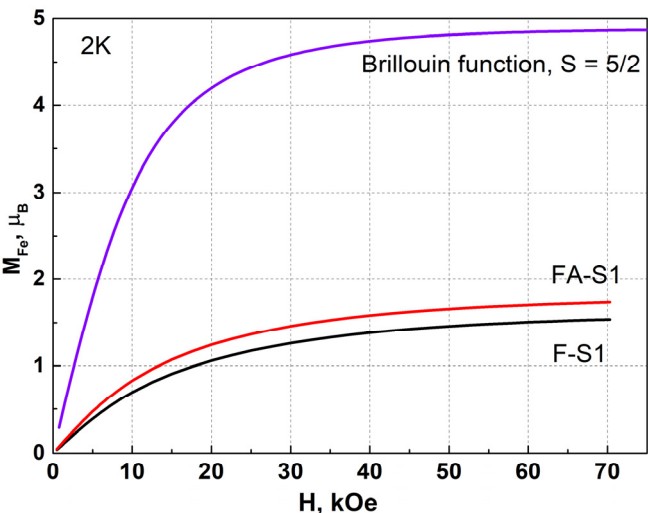

**Figure 6.** Magnetization curves of F-S1-1 and FA-S1-3 samples at 2 K.

The temperature dependence of the magnetic susceptibility of samples F-S1 and FA-S1 is shown in Figure 7. Extrapolation from the temperature range (150–300) K gives the values of the Weiss constant $\theta = -45$ K and $\theta = -39$ K for samples F-S1-1 and FA-S1-3, respectively. Thus, from a comparison of the magnetization curves and the temperature dependences of the susceptibilities, it follows that the addition of aluminum leads to an insignificant change in the negative exchange interactions in iron clusters, decreasing them in absolute value. This slight change, however, probably indicates that aluminum ions prefer to accumulate near iron nodes.

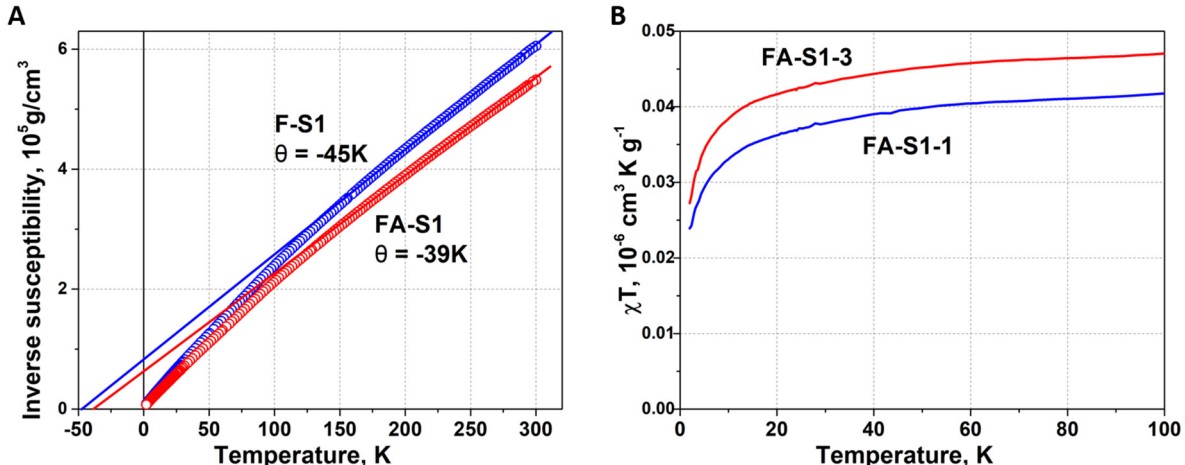

**Figure 7.** (**A**): Temperature dependence of the inverse susceptibility for samples F-S1 and FA-S1. H = 1 kOe. Extrapolation was done from the high-temperature region (150–300 K). (**B**): Temperature dependence of the value ($\chi \cdot T$) for samples F-S1 and FA-S1. H = 1 kOe.

A more in-depth interpretation of the temperature dependences of the susceptibility is also possible. In [31], typical dependences of the parameter ($\chi \cdot T$) versus T are given for various variants of interaction in dimers and trimers. The experimental dependences of this parameter for samples FS-1-1 and FA-S1-3 are shown in Figure 7B. The experimental dependences of this parameter for samples FS1-1 and FAS1-3 are shown in Figure 7B. They have the form of increasing functions, which is typical for clusters with negative interaction. For dimers at low temperatures, the parameter should tend to zero, and for clusters (in particular, trimers for any combination of signs of interaction), the value ($\chi \cdot T$) tends to a nonzero value with decreasing temperature [31]. The observed form of this dependence is more in line with trimers or, rather, clusters with a set of negative exchange interactions.

On Figure 8A shows the magnetization curves of the samples obtained by the second chemical route.

**Figure 8.** (**A**): Magnetization curves for samples synthesized by the second route: 1- F-S2-1-300, 2- FA-S2-3-300 (X-ray amorphous state). (**B**): Temperature dependence of the inverse susceptibility for samples FA-S2-3-300 and F-S2-1-300. H = 1 kOe. Extrapolation was done from the high-temperature region (150–300 K).

The magnetization curves for the samples with Al and without Al, as well as for the samples of the first series, differ little, indicating that the main negative exchange interactions are largely retained. This is also evidenced by a comparison of the Weiss constants for samples with and without aluminum (see Figure 8B).

It should be noted that the signs of the Weiss constants remain negative, emphasizing the preservation of negative exchange interactions. It is known that the absolute value of Theta is not a measure of the magnitude of exchange interactions in the system. It can only be noted that its modulus is relatively weakly dependent on the aluminum content in the $TiO_2$:Fe, Al system.

Thus, in the samples synthesized both by the first and the second route, regardless of their structural state, aluminum, considering the accuracy of determining the chemical composition, practically does not affect the magnitude of the magnetic moment of the iron ion and the magnitude of negative exchange interactions within clusters gland.

### 3.5. Analysis of the EPR Spectra of Samples at Room Temperature

The EPR spectra of samples FS1 and FAS1 are shown in Figure 9. The addition of aluminum leads to some decrease in the intensity of the line with a g factor of 2 (nondistorted octahedral environments) and a significant increase in the intensity of the line with a g factor of 4.3 (distorted octahedral or tetrahedral environments). This effect is even more pronounced on samples obtained by the CR2 method, calcined at 300 °C (Figure 10). This redistribution of EPR line intensities may be the result of either the displacement of $Fe^{3+}$ ions by $Al^{3+}$ ions because of substitutional occupation, or the consequence of the distortion of the nearest environment of $Fe^{3+}$ ions due to interstitial Al. Since calculations have shown that aluminum ions cannot replace iron ions, we believe that the second option is being realized. In this case, as noted above, the magnetic properties, for example the magnetization and the Weiss constant, change insignificantly when doped with aluminum.

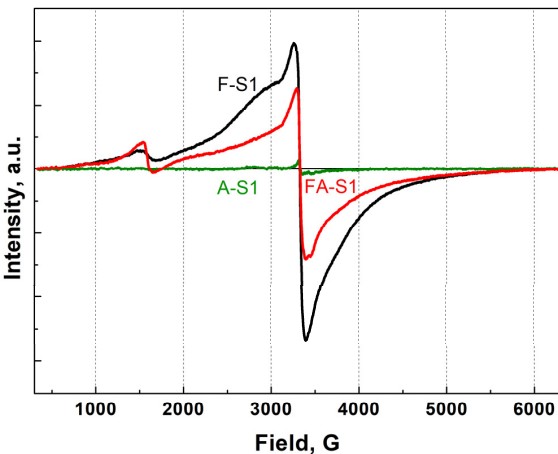

**Figure 9.** EPR spectra of samples F-S1 (black line), A-S1 (green line) and FA-S1 (red line) at RT.

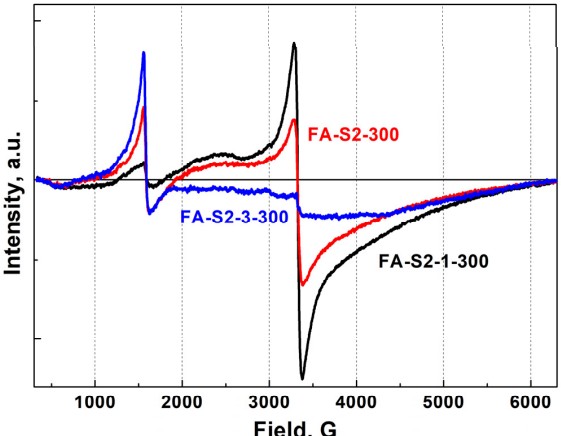

**Figure 10.** EPR spectra of samples FA-S2-1-300 (black line), FA-S2-2-300 (red line) and FA-S2-3-300 (blue line) at RT.

Additional changes in the signal with g = 4.3 in the samples prepared by the first route with varying Al content are a convincing demonstration that Al ions are close to iron ions. This is also observed for samples calcined at 500 °C, although the addition of aluminum affects the ratio of line intensities much less (Figure 11) than for samples calcined at lower temperatures. It is possible that, to a significant extent, this may be due to the coarsening of nanoparticles and a decrease in the contribution of surface states.

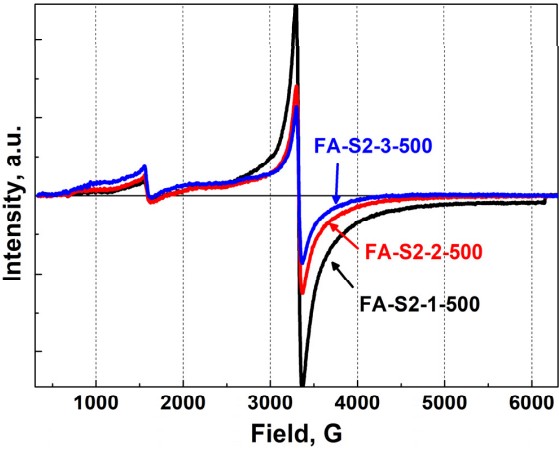

**Figure 11.** EPR spectra of samples FA-S2-1-500 (black line), FA-S2-2-500 (red line) and FA-S2-3-500 (blue line) at RT.

In this regard, of fundamental importance for the nano state is the localization of distorted octahedral or tetrahedral distortions on the surface or in the volume of nanoparticles. Indeed, in several studies, the appearance of a signal with a g factor of 4.3 is associated mainly with distortions of the nearest environment near iron ions, which can be localized mainly on the surface of nanoparticles [32–40]. Cases of the X-ray amorphous state were also considered in the study using the EPR method [32], where it was shown that the signal g = 4.3 prevails in comparison with the signal g = 2.

The X-ray amorphous state is the case when distortions in the nearest environment can be observed throughout the volume of the amorphous body and, obviously, the signal with g = 4.3 will dominate (see Figure 10). The transition from an amorphous state to a crystalline state, either because of a change in the synthesis conditions or a change in composition, and calcination at higher temperatures, is accompanied by an increase in the perfection of the lattice and the growth of nanoparticles. As a result, one should expect a decrease in the contribution of the imperfect lattice and a simultaneous increase in the surface contribution at small nanoparticle sizes. It is obvious that the surface always remains a place of localization of various defects and broken bonds, including coordination disorders. As the size of nanoparticles increases, the proportion of the surface contribution decreases and, therefore, the signal with g = 4.3 will decrease compared to g = 2. Of course, one cannot exclude the presence of rhombohedral distortions near $Fe^{3+}$ ions due to defects (oxygen vacancies, interstitial atoms, etc.) near these sites in the crystalline state, which can make a relatively small contribution due to a perfect lattice, which is shown in the EPR spectra in Figures 9 and 11. Of course, in this case, a small fraction of distorted surface contributions with g = 4.3 in the EPR spectra is also retained.

So, qualitatively, the behavior of the EPR spectra (the presence of signals g = 4.3 and g = 2 from $Fe^{3+}$ ions) with different amplitudes can be described as a changing ratio of the proportion of the amorphous or distorted structural state in the sample volume and the proportion of distorted states, localized, mainly, on the surface of more perfect nanoparticles after various calcination modes. In this case, it should be emphasized that, considering the clustering of iron ions and the influence of the dipole–dipole and exchange interactions between Fe ions, the signal with g~2 can have a more complex shape [26,30,41]. This is also clearly demonstrated in the present work for $TiO_2$:Fe and Al samples, regardless of the synthesis method.

*3.6. Temperature Dependences of the Components of the EPR Spectra*

As noted in [31], the initial susceptibility of clusters (dimers or trimers) in coordinates ($\chi \cdot T$) vs. T depends on the sign of the interaction. In the absence of interaction, of course, the value ($\chi \cdot T$) does not depend on T in accordance with the Curie law. For dimers with positive interaction, a decrease in the value of ($\chi \cdot T$) should be observed with a plateau corresponding to the Curie law. For dimers with a negative interaction, growth should be observed at low T, and the beginning of the growth region is shifted along the temperature axis. For trimers (probably, rather clusters), with all possible variations of negative interactions, growth also takes place, but extrapolation to zero temperatures should lead to an intersection with the ($\chi \cdot T$) axis and not with the T axis, as in the case of dimers with negative interaction.

The area under the individual peaks of the EPR spectra after double integration is proportional to the initial susceptibility. Then the product of area and temperature should behave in the same way as the product of initial susceptibility and temperature. We measured the EPR spectra at low temperatures (from 8 to 90 K) for three samples with different Al contents in order to check whether the addition of Al has any effect on the susceptibility of those iron locations in which aluminum atoms are present in the nearest vicinity. The EPR spectra for one of the samples are shown in Figure 12. The intensity of the lines increases strongly with decreasing temperature, which is quite natural for states without long-range magnetic order. To analyze the temperature dependence of the intensity of individual lines, the spectra were first integrated and then decomposed into three separate Gaussian lines.

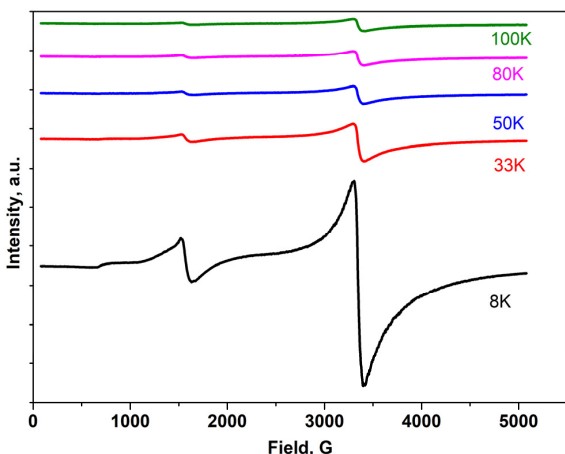

**Figure 12.** EPR spectra of sample FA-S2-1-300 in the range from 8 to 100 K.

One line corresponds to a g-factor of 4.3; two other lines of different widths correspond to signals with a g-factor close to 2. An example of such a decomposition is shown in Figure 13.

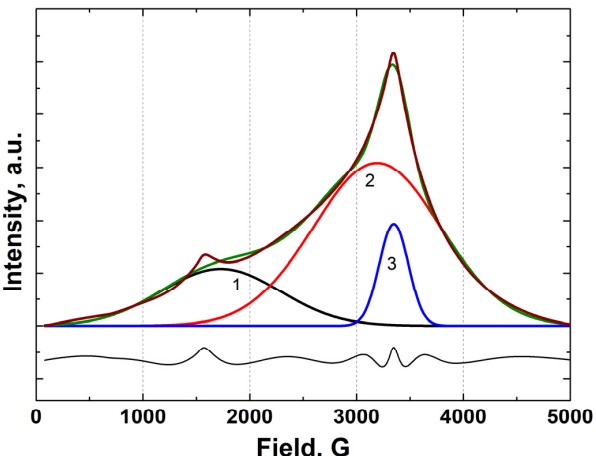

**Figure 13.** Deconvolution of the integral EPR spectrum of the FA-S2-1-300 sample (measured at 8 K) into three spectral lines. Black line—the reduced error of the decomposition.

Next, the values of the integral intensity of each line I were obtained, and the values of the parameter (I·T) were calculated for different Ts in the range (8–90) K (Figure 14).

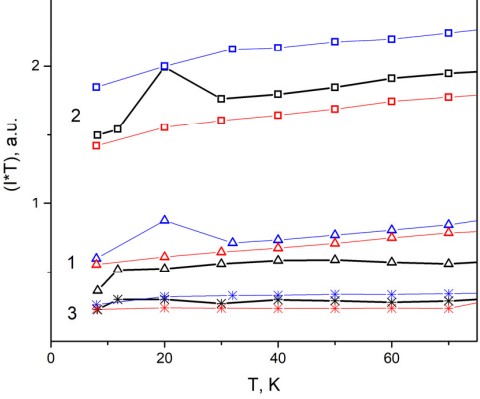

**Figure 14.** Temperature dependences of intensity of the separate resonance lines: line 1 (triangles), 2 (squares) and 3 (asterisk) for samples F-S-1, FA-S2-1-300, FA-S2-2-300 with different Al content: 0 at.% (black), 0.04 at.% (blue) and 0.81 at.% (red)) respectively.

The intensity of the lines corresponding to the noninteracting moments of the iron ions (3) behaves in accordance with the Curie law, and the intensity of the lines with a g factor of 4.3 (1) also manifests itself in a similar way. The parameter (I·T) for the lines corresponding to the weakly interacting moments of iron (curves 2) shows some increase with temperature, which is typical for clusters [31]. To some extent, this behavior correlates with the behavior of the parameter (χ·T) (Figure 7B), which serves as additional confirmation in favor of the realization of, not dimers, but of trimers or more complex iron clusters. However, strictly speaking, for a more reliable proof of this statement, it is necessary to measure both the susceptibility and the EPR spectra in the temperature range even closer to zero.

The main conclusion that can be drawn from the analysis of the temperature dependence of the EPR spectra is that the local order of the nearest environment of iron ions (the presence of aluminum ions there or their absence) does not affect the change in individual components of the spectra with temperature.

## 4. Conclusions

We used two sol-gel methods for the synthesis of $TiO_2$:Fe, Al samples. The first option allows one to obtain nanoparticles (with an average size of about 20 nm) with an anatase crystal structure and, in the second one, more dispersed nanocrystals (less than 5 nm) can be synthesized, including the X-ray amorphous state after calcination at 300 °C.

Calcination of the samples at 400–500 °C synthesized using the second method leads to the growth of nanoparticles (~20 nm). The properties of nanoparticles received by both synthesis methods are similar.

The first-principle calculation of the localization of aluminum in the anatase lattice made it possible to establish that $Al^{3+}$ is in the interstitials of the $TiO_2$ lattice. This localization of aluminum mainly leads to lattice distortions near $Fe^{3+}$ ions and does not significantly change the magnetic properties of titanium dioxide doped with Fe and Al.

These distortions are recorded when analyzing the EPR spectra. With an increase in the aluminum content in titanium dioxide with iron, the ratio of the signal's intensity in the EPR spectra with g factors of 2 and 4.3 will noticeably change. The ratio between signals 4.3 and 2 also depends on the fraction of the amorphous phase and the contribution of the surface states of the nanoparticles.

The deconvolution of the EPR spectrum of $TiO_2$:Fe samples with and without Al into separate spectral components has been carried out. It is shown that the experimental spectrum can be fitted by three separate spectral Gaussian lines. The analysis of the temperature dependence Intensity * T(T) for different lines made it possible to establish that these dependences can be qualitatively described in the model of clusters with negative exchange interactions between iron ions.

Thus, the magnetic properties of the samples change insignificantly on doping with aluminum. The main magnetic interactions are retained in this system with a set of individual paramagnetic centers and the presence of clusters of iron ions. The negative exchange interactions remain almost unchanged upon doping with Al.

**Author Contributions:** Conceptualization, A.Y.; methodology, A.Y., M.U. and A.M.; validation, A.Y. and M.U.; investigation, M.U., K.B., A.M., D.B., D.S., A.V., R.E., I.Y., G.Z. and V.G.; writing—original draft preparation, A.Y. and M.U. All authors have read and agreed to the published version of the manuscript.

**Funding:** This work is financially supported by the RFBR Grant #20-02-00095 and partly by the Ministry of Science and Higher Education of the Russian Federation (theme "Magnet", project No. 122021000034-9). Electron spin resonance measurements (Rushana Eremina, Ivan Yatsyk) were performed with the financial support from the government assignment for FRC Kazan Scientific Center of RAS.

**Institutional Review Board Statement:** Not applicable.

**Informed Consent Statement:** Not applicable.

**Data Availability Statement:** Not applicable.

**Acknowledgments:** X-ray diffraction and transmission electron microscopy investigations were performed in the Collaborative Access Center of the M.N. Mikheev Institute of Metal Physics of the Ural Branch of the Russian Academy of Sciences. The authors are grateful for the contributions made by our colleagues M.I. Kurkin, A.F. Gubkin, A.V. Korolev, I.A. Kurmachev, A.S. Konev.

**Conflicts of Interest:** The authors declare no conflict of interest.

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
