# Peer review of "Magnetism and EPR Spectroscopy of Nanocrystalline and Amorphous TiO2: Fe upon Al Doping"

_magnetochemistry, doi:10.3390/magnetochemistry9010026_

Round 1

Reviewer 1 Report

 The manuscript Magnetism and EPR spectroscopy of nanocrystalline and amorphous TiO2: Fe upon Al dopingpresents the results of materials research using the EPR method.  The characteristics of the crystal lattice under the influence of aluminum and iron impurities were studied. The contribution of Fe3+ and Al3+ ions to the magnetic properties of titanium dioxide was determined. Distortion of the crystal lattice was analyzed by EPR spectra. A change in the ratio of signal intensities in EPR spectra was detected. In addition, the ratio between the signals depends on the proportion of the amorphous phase and the contribution of nanoparticles.

Notes for correction:

35-  It would be possible to unify the fonts in the notation (index 2): TiO2  35,43….  and  TiO2   39,78, 107…..  

92- Definition electron density functional (DFT) - Is the abbreviation the same?  see169

169- Definition density functional theory  (DFT) - Is the abbreviation the same? see 92

240- the indices of the reference crystal planes could be inserted in the Figure 3

396-along the Y axis is omitted “Intetsity, a.u.”- Figure 12

General characteristics of the manuscript:

             The work is written clearly and distinctly.

             Graphs and formulas are clear and informative.

             The work can be published in the presented version, after minor corrections.

Conclusion:

The work can be published in the presented version, after minor corrections.

Author Response

35-  It would be possible to unify the fonts in the notation (index 2): TiO2  35,43….  and  TiO2   39,78, 107…..  

All indices in the "TiO2" formula have been subscripted.

92- Definition “electron density functional (DFT)” - Is the abbreviation the same?  see169

169- Definition “density functional theory  (DFT)” - Is the abbreviation the same? see 92

Yes, the same abbreviation and definition are used.

240- the indices of the reference crystal planes could be inserted in the Figure 3

Anatase lattice indices added to figure 3.

396-along the Y axis is omitted “Intetsity, a.u.”- Figure 12

Axis title added

Reviewer 2 Report

This paper reports the experimental and density functional theory (DFT) calculation of Al and Fe-doped TiO2. Doped TiO2 is a ubiquitous material with plenty of catalytic, bio-, and magnetic applications. With the following revisions, the paper may be potentially considered for publication:

1.       The chemical formula in the first column of Table 1 should reflect the materials more accurately. Consider TiO2Al à TiO2:Al.

2.       The procedure for calculating the DFT formation energy should be explained in the methods.

3.       The GGA functional, used in the DFT calculations, systematically underestimates the band gap in oxides. The authors must elaborate that their calculations are quantitative in nature.

4.       Is it viable that the authors perform complete refinement on their XRD observation for the crystalline phases? If not, the reasons should be explained.

5.       DFT calculations for Al and Fe-doped TiO2 have been previously reported in the literature. A comparison with these results should be given in the discussions:

Hanaor et al. Ab initio study of phase stability in doped TiO2 Computational Mechanics volume 50, pages185–194 (2012); https://doi.org/10.1007/s00466-012-0728-4

Author Response

 1. The chemical formula in the first column of Table 1 should reflect the materials more accurately. Consider TiO2Al à TiO2:Al.

Chemical formula in the table 1 have been changed to be more correct e.g. TiO2:Al

2. The procedure for calculating the DFT formation energy should be explained in the methods.

Answer. Thank you. The formula for the calculation of formation energy have been added in revisited
manuscript. The lines from 179 to 187.

3. The GGA functional, used in the DFT calculations, systematically underestimates the band gap in oxides.

The authors must elaborate that their calculations are quantitative in nature.
Answer. Yes, this is correct, standard DFT underestimate the value of the bandgap in all material. In our manuscript we do not discuss neither electrical nor optical properties of considered systems and hence underestimation of the bandgap width does not influence simulated properties such as energetics of defects formation, values of magnetic moments and exchange interactions.

4. Is it viable that the authors perform complete refinement on their XRD observation for the crystalline phases?
If not, the reasons should be explained.

Answer. A Rietveld refinement of the XRD data was carried out where possible. Some of the samples (namely FA-S2-2-300 and FA-S2-3-300) probably have a too wide particle size distribution, which makes it impossible to process them according to the Rietveld. The processing results are added to the
corresponding graphs along with the difference data.

5.       DFT calculations for Al and Fe-doped TiO 2  have been previously reported in the literature. A comparison with these results should be given in the discussions:

Hanaor et al. Ab initio study of phase stability in doped TiO2 Computational Mechanics volume 50,
pages185–194 (2012); https://doi.org/10.1007/s00466-012-0728-4

Answer. In mentioned work had been considered only Al-doped TiO2 and Fe-doped TiO2. Simultaneous
doping TiO2 by Al and Fe had not been considered in the paper pointed by the reviewer. Additionally, in
mentioned paper smaller supercell (72 atoms) was used. In the work reported in our manuscript the supercell
of 96 atoms have been used. We added citation of the work pointed by the reviewer in revisited manuscript
(ref. 24).

Round 2

Reviewer 2 Report

In light of the revisions performed, the paper can be accepted in the current form.